# Lonidamine Induced Selective Acidification and De-Energization of Prostate Cancer Xenografts: Enhanced Tumor Response to Radiation Therapy

**DOI:** 10.3390/cancers16071384

**Published:** 2024-03-31

**Authors:** Stepan Orlovskiy, Pradeep Kumar Gupta, Jeffrey Roman, Fernando Arias-Mendoza, David S. Nelson, Cameron J. Koch, Vivek Narayan, Mary E. Putt, Kavindra Nath

**Affiliations:** 1Molecular Imaging Laboratory, Department of Radiology, University of Pennsylvania, Philadelphia, PA 19104, USA; stepan.orlovskiy@pennmedicine.upenn.edu (S.O.); pradeep.gupta@pennmedicine.upenn.edu (P.K.G.); rjeff@pennmedicine.upenn.edu (J.R.); fernando.arias-mendoza@pennmedicine.upenn.edu (F.A.-M.); dsnelson@pennmedicine.upenn.edu (D.S.N.); 2Advanced Imaging Research, Inc., Cleveland, OH 44114, USA; 3Department of Radiation Oncology, University of Pennsylvania, Philadelphia, PA 19104, USA; kochc@pennmedicine.upenn.edu; 4Department of Medicine, University of Pennsylvania, Philadelphia, PA 19104, USA; vnarayan@pennmedicine.upenn.edu; 5Department of Biostatistics and Epidemiology, University of Pennsylvania, Philadelphia, PA 19104, USA; mputt@pennmedicine.upenn.edu

**Keywords:** lonidamine, prostate cancer, radiation therapy, tumor acidification, tumor de-energization, ^1^H and ^31^P magnetic resonance spectroscopy, radiation growth delay

## Abstract

**Simple Summary:**

Early prostate cancer treatment with radiation therapy is effective but involves substantial harmful side effects throughout therapy. This study explores the use of the metabolic modulator lonidamine (LND) to sensitize prostate cancer cells to radiation therapy. ^1^H and ^31^P magnetic resonance spectroscopy (MRS) were used to non-invasively monitor metabolic changes associated with LND treatment in human prostate cancer tumors implanted in athymic nude mice. The in vivo MRS results were substantiated using classic biochemistry methods by gathering in vitro data from isolated human prostate cancer cell lines. A radiation growth delay experiment performed in xenografted mice combining LND and radiation therapy showed that LND-modified prostate cancer metabolism and significantly increased the efficacy of radiation therapy. These findings indicate that less radiation may be required with LND, reducing side effects, and potentially improving patient management in the clinic.

**Abstract:**

Prostate cancer is a multi-focal disease that can be treated using surgery, radiation, androgen deprivation, and chemotherapy, depending on its presentation. Standard dose-escalated radiation therapy (RT) in the range of 70–80 Gray (GY) is a standard treatment option for prostate cancer. It could be used at different phases of the disease (e.g., as the only primary treatment when the cancer is confined to the prostate gland, combined with other therapies, or as an adjuvant treatment after surgery). Unfortunately, RT for prostate cancer is associated with gastro-intestinal and genitourinary toxicity. We have previously reported that the metabolic modulator lonidamine (LND) produces cancer sensitization through tumor acidification and de-energization in diverse neoplasms. We hypothesized that LND could allow lower RT doses by producing the same effect in prostate cancer, thus reducing the detrimental side effects associated with RT. Using the Seahorse XFe96 and YSI 2300 Stat Plus analyzers, we corroborated the expected LND-induced intracellular acidification and de-energization of isolated human prostate cancer cells using the PC3 cell line. These results were substantiated by non-invasive ^31^P magnetic resonance spectroscopy (MRS), studying PC3 prostate cancer xenografts treated with LND (100 mg/kg, i.p.). In addition, we found that LND significantly increased tumor lactate levels in the xenografts using ^1^H MRS non-invasively. Subsequently, LND was combined with radiation therapy in a growth delay experiment, where we found that 150 µM LND followed by 4 GY RT produced a significant growth delay in PC3 prostate cancer xenografts, compared to either control, LND, or RT alone. We conclude that the metabolic modulator LND radio-sensitizes experimental prostate cancer models, allowing the use of lower radiation doses and diminishing the potential side effects of RT. These results suggest the possible clinical translation of LND as a radio-sensitizer in patients with prostate cancer.

## 1. Introduction

Prostate cancer is the second most common cancer in men and the second leading cause of male cancer death [1,2]. There are more than 10 million people living with prostate cancer globally, with over 1.4 million new cases being diagnosed every year and 375,304 reported deaths from the disease [3]. In the United States in 2023, there were 288,300 reported cases of prostate cancer, with an estimated 34,700 deaths, and this number is expected to increase in the future [4]. Treatment of prostate cancer depends on the prognosis as well as patient preference, physician recommendation, operability, and comorbidity [5]. Indolent prostate cancer is often treated through active surveillance to preserve bodily function. In contrast, higher-risk localized prostate cancers are treated with ablative radiation therapy (RT) or radical prostatectomy with curative intent [6]. Overall, studies show that curative treatment might be beneficial in all but the most indolent (or advanced) prostate cancers [7,8]. Men with local or systemic relapsing disease are treated with androgen deprivation therapy (ADT), either alone or in combination with RT or chemotherapy [3,6]. Although initial responses to these therapies are favorable, 10–20% of cases progress to metastatic castration-resistant prostate cancer (mCRPC), which is refractory to ADT and has poor survival [9,10]. At this advanced stage, the disease becomes incurable, and chemotherapy becomes the standard of care to slow its progression [7,8,10,11].

External Beam Radiation Therapy (EBRT) and Brachytherapy (BT) are the most common radio-therapeutic techniques used in the treatment of prostate cancer. EBRT delivers ionizing radiation generated by an external radiation source in effective doses of 50–80 Gray (GY), improving overall survival for most patients [12]. The therapeutic radiation dose delivered in EBRT therapy is split into fractions in a process called conventional fractionation to lessen the toxicity in normal tissues associated with a single large dose of radiation; the therapy is performed over 8–9 weeks and is delivered in 39–45 fractions of 1.8 GY to 2 GY per fraction [12]. Hypo-fractionated EBRT uses fewer higher-dose fractions (19–28) of 2.5 GY to 3.4 GY per fraction for similar but potentially improved cancer control compared to the standard [12]. BT is a form of treatment where radiation is delivered internally through surgically implanted radioactive seeds or capsules, permitting the delivery of a more significant radiation dose than other RT treatments [12]. In Low Dose Rate BT, radioactive seeds are permanently implanted into the prostate, delivering an average dose of 0.1 GY/h for several months, with an effective total dose of 120–160 GY depending on the type of seed used and the specific patient situation and tumor status [12]. In High-Dose Rate BT, temporary catheters are inserted in the prostate, where radiation is delivered in greater doses of 27–35 GY in several fractions [13]. A combination of EBRT and Low Dose Rate BT can also be used for high risk patients in situations where the cancer has progressed from the prostate into the surrounding pelvic lymph nodes [14].

Novel applications of these established radio-therapeutic techniques and new treatment modalities seek to reduce the risk of toxic effects of radiation on healthy tissue around the targeted tumor, but treatment-associated toxicity is still prevalent. RT causes urinary dysfunction, leading to irritative and obstructive symptoms, like urinary frequency, urgency, dysuria, urethral obstruction, hematuria, and bladder contracture [15]. Sexual function also declines as RT causes decreased potency through direct damage to nerves and arterial/venous insufficiency [15]. Gastro-intestinal dysfunction also occurs with rectal urgency, rectal bleeding, cramping, diarrhea, proctitis, and incontinence [15]. Rarely, EBRT can induce a second cancer in irradiated tissues. Thus, the risk of bladder or rectal cancer is a consideration when treating younger men with prostate cancer [15].

Based on our work with other cancer models (e.g., melanoma, breast cancer), refs. [16,17,18] our goal was to determine if the metabolic modulator lonidamine (LND) exerts a similar effect in prostate cancer and potentiates the impact of radiation therapy in low-risk prostate cancer. If LND could provide radio-sensitization, it could allow for the lowering of the radiation dose, diminishing the harmful side effects associated with RT while effectively treating organ-confined disease. The LND effect is based on the metabolic reprogramming of cancer cells, tending to increase glucose uptake and lactate production by shifting their metabolism to upregulate glycolysis (i.e., Warburg Effect) [19]. This cancer metabolic property can be exploited by using the metabolic modulation of LND, causing cancer-selective intracellular acidification by trapping the lactic acid inside the cell via inhibition of the monocarboxylate transporters (MCT1 and MCT4), inducing cell death [20]. Additionally, LND damages cancer cells by inhibiting the mitochondrial pyruvate carrier (MPC) and complex II of the electron transport chain [20,21]. LND also decreases levels of the metabolic reducing agent glutathione (GSH) and induces reactive oxygen species (ROS) formation by increasing tumor oxygen concentration, making cells more vulnerable to treatment, including RT [21]. As a result of the intracellular acidification LND induces in cancer cells, it could also inhibit hexokinase-2 and other glycolytic enzymes and steps of the pentose phosphate pathway [21,22]. Furthermore, LND has been proven safe in clinical trials of cancers as it lacks toxicities to bone marrow and other tissues. However, it is not devoid of side effects, like myalgia and testicular pain [23,24].

Concurrently, we have also shown that non-invasive metabolic imaging methods based on magnetic resonance spectroscopy (MRS) provide reliable in vivo metabolic biomarkers that could potentially be translated for a personalized clinical approach. In the case of LND, measuring lactate and bioenergetics by ^1^H- and ^31^P-MRS could identify the optimal time window for treatments, precisely the time frame of maximum LND-induced acidification. The personalized approach can also identify those patients who are responsive to LND treatment while sparing non-responders. Our approach, if successful, has the potential to enhance the well-being of men experiencing organ-confined prostate cancer.

## 2. Materials and Methods

### 2.1. Reagents and Mice

LND and 3-aminopropylphosphonate (3-APP) were purchased from Santa Cruz Biotechnology, Inc. (Santa Cruz, CA, USA). RPMI 1640 medium, glutamine, HEPES, penicillin-streptomycin, trypsin-EDTA, DPBS, sodium pyruvate, and HBSS were purchased from Thermo Fisher Scientific (Waltham, MA, USA; Invitrogen brand). FBS was purchased from HyClone Laboratories, Inc. (Logan, UT, USA). Glucose, DMSO, trizma base, and glycine were purchased from Sigma-Aldrich (St. Louis, MO, USA). T-75 flasks, trypan blue, and 10-cm culture dishes were purchased from Corning Life Sciences (Tewksbury, MA, USA; Falcon brand). Seahorse XF Base DMEM Medium, oligomycin, FCCP, rotenone, and antimycin A were purchased from Seahorse Bioscience, Inc. (North Billerica, MA, USA). 

Animal studies were performed under the approval of the University of Pennsylvania Institutional Animal Care and Use Committee (IACUC). Male athymic nude mice (01B74), 4–6 weeks of age, were purchased from the National Cancer Institute (Frederick, MD, USA) and housed in microisolator cages with access to water and autoclaved mouse chow ad libitum. For in vivo MRS and growth delay experiments, the following accessories were used: a respiration pillow and rectal probe from SA Instruments Inc. (Stony Brook, NY, USA; Model 1025), water pad heater from Gaymar Industries, Inc. (Orchard Park, NJ, USA), catheters from Tyco Healthcare (Milwaukee, WI, USA), a restrainer from Braintree Scientific, Inc. (Braintree, MA, USA), calipers from Bel-Art Products (Wayne, NJ, USA), and scale from H and C Weighing Systems (Columbia, MD, USA; Acculab PP401).

### 2.2. Prostate Cell Line Culture

PC3 prostate cancer cells were purchased from ATCC (Manassas, VA, USA). They were cultured in RPMI 1640 medium, which was supplemented with fetal bovine serum (10%), sodium pyruvate (1%), and penicillin-streptomycin (1% *V*/*V*, 100 U/mL final). Cells were grown in this supplemented culture medium in T-75 or T-150 flasks and incubated at 37 °C and 5% CO_2_. Cells were passaged at ~90% confluence; medium was removed from flasks, the flasks washed with Hanks Balanced Salt Solution, and cells incubated in cold 0.05% trypsin-EDTA for 9–10 min before being split (1:16) into new flasks. Cells were counted with a hemocytometer and trypan blue stain when specific cell numbers were needed.

### 2.3. Seahorse and YSI Cell Metabolism Assays

An XFe96 Seahorse Extracellular Flux Analyzer was used to measure the oxygen consumption rate (OCR) and extracellular acidification rate (ECAR) in vitro. The assay medium was formulated to match the culture medium by supplementing Seahorse XF Base Medium with 5 mM glucose, 2 mM L-glutamine, and 1 mM sodium pyruvate. 2 × 10^4^ PC3 cells were seeded (per well) into an XF96 cell culture microplate. In our protocol, three basal OCR and ECAR measurements were taken in a 3–0-3 mix-wait-measure cycle. After basal measurements, LND (0, 1, 10, 100, 150, or 200 µM; 0.2% DMSO) was injected into the wells, and measures were taken for the next three hours. Then, 0.5 µM FCCP and 1 µM oligomycin were injected into the wells, and three measurements were taken. Lastly, 0.5 µM rotenone and 0.5 µM antimycin A were added, and three measurements were taken. The cells were proteolyzed with 0.05% trypsin-EDTA and counted with a hemocytometer.

Glucose consumption and lactate production rates of the cells in media were determined using a YSI 2300 STAT Plus glucose and lactate analyzer calibrated with YSI 2747 dual standard. PC3 cells were seeded in 6-well plates at a density of 1 × 10^6^ cells in 2 mL of RPMI 1640 growth medium (0.2% DMSO ± 150 μM LND) per well. Extracellular glucose and lactate concentrations were measured at 24- and 48-h post-treatment intervals.

### 2.4. Human Prostate Cancer Xenografts in Athymic Nude Mice

PC3 prostate cancer xenografts were grown in athymic nude mice. Briefly, 7 × 10^6^ cells in 100 µL HBSS were inoculated subcutaneously in the right flank of nude mice. In vivo xenograft doubling times were calculated by measuring tumor growth in untreated mice with a caliper, using the formula for the volume of a half-ellipsoid, *V* = (π/6) (*l* × *w* × *d*), where *l*, *w*, and *d* are the tumor’s length, width, and depth, respectively.

### 2.5. Non-invasive Magnetic Resonance Spectroscopy (MRS) Measurements

Non-invasive MRS experiments were performed in the tumors of PC3 mouse xenografts. The tumors were hemispherical with a volume of approximately 250 mm^3^. For these experiments, we used a 9.4 T/31-cm horizontal bore Varian system (Varian Medical Systems, Inc., Palo Alto, CA, USA). Before the MRS measurements, tumor-bearing mice were anesthetized with 1% isoflurane in 1 L/min oxygen flow, and two 26-gauge catheters were inserted into either side of the peritoneum. Respiration rate and core body temperature were monitored using a respiration pillow and a rectal thermistor. Core body temperature was maintained at 37 ± 1 °C by blowing warmed air into the magnet’s bore with a thermal regulator system.

To non-invasively determine pH and bioenergetic status, ^31^P MRS was acquired by positioning the tumor-bearing mouse in a built-in-house, dual-frequency (^1^H/^31^P) slotted-tube resonator (10 mm diameter) after the animal was injected with 3-APP (0.2 mL of a 300 mg/mL). The Image Selected In vivo Spectroscopy (ISIS) technique was used with the following parameters: Hyperbolic Secant-Adiabatic Fast Passage (HS-AFP) slice-selective inversion pulses with 2.5 ms length, 296 scans with a radiofrequency pulse width of 60 µs, corresponding approximately to a 90° flip angle; 12 kHz sweep width; 512 data points; TR = 4 s. The in vivo pH changes were determined by measuring the chemical shift of the inorganic phosphate (Pi) signal for intracellular pH (pHi) and that of 3-APP for extracellular pH (pHe) in the ^31^P spectrum. Bioenergetics was assessed by integrating the β-signal of nucleoside triphosphates and dividing it by the integral of the Pi signal (i.e., βNTP/Pi).

Lactate was measured using ^1^H MRS by positioning the tumor in a built-in-house, single frequency (^1^H) slotted-tube resonator (13 mm inner diameter, 15 mm outer diameter, 16.5 mm depth). A slice-selective, double-frequency, Hadamard-selective, multiple quantum coherence (HDMD-Sel-MQC) transfer pulse sequence was used to edit the ^1^H methyl signal from lactate in the ^1^H MRS spectrum, filtering out the overlapping lipid signals [25]. The following acquisition parameters were used: sweep width = 4 kHz, 2048 data points, TR = 8 s, and 128 scans. The sequence also includes water suppression pulses.

The magnet was shimmed globally until the tumor water signal monitored via the ^1^H channel reached 60–70 Hz of linewidth at half height in all MRS experiments. The Point Resolved Spectroscopy (PRESS) sequence was used to shim the ISIS (^31^P MRS) or HDMD-Sel MQC-selected (^1^H MRS) tumor voxel to a water signal linewidth at half height of 30–40 Hz. The tumor voxel volume was 250–300 mm^3,^ covering as much tumor volume as possible.

After acquiring baseline MR spectra, LND (100 mg/kg) was administered in the experimental animals or its vehicle solution in the control animals through the intraperitoneal (i.p.) catheters without moving the animals from the magnet. The vehicle solution was a buffer consisting of trizma base (1.2 g) and glycine (5.76 g) in sterile water (100 mL; final pH = 8.3). A LND solution (22.0 mg/mL in the tris/glycine buffer) was prepared and spun at 40 °C until completely dissolved.

NUTS (Acorn NMR Inc., Livermore, CA, USA) and MestRec (Mestrelab Research, Santiago de Compostela, Spain) were used to process all spectroscopic data. A 20 Hz exponential filter was used to improve the apparent signal-to-noise ratio of ^31^P MRS data, and baseline correction was applied before plotting and integrating peak areas. pHi and pHe were determined from the Henderson-Hasselbach equation using the chemical shifts of Pi and 3-APP, respectively, referenced to αNTP resonances. For pHe, the pKa of 6.90 ± 0.03, limiting acid chemical shift of 34.22 ± 0.04 ppm, and base chemical shift of 31.08 ± 0.04 ppm were used, and for pHi, the corresponding parameters were 6.57 ± 0.03, 13.52 ± 0.03, and 11.24 ± 0.02 ppm, respectively [17,26,27]. As previously described, the βNTP/Pi ratio determined the bioenergetic status. Although the NTP signal includes all the nucleoside triphosphates, we considered it representative of the intracellular ATP (adenosine triphosphate) levels. This is because the ATP content is the highest NTP in the cell, and the transfer of the terminal phosphorous between NTPs is in equilibrium. In addition, we used the βNTP signal as it does not overlap with other spectral signals. Lactate was integrated into the water-suppressed HDMD-Sel-MQC spectra, and the water signal (H_2_O) was integrated into the non-water-suppressed spectra. The Lactate/H_2_O was determined to represent the intracellular lactate molar concentration [28].

### 2.6. Treatment with Radiation and Growth Delay Measurements

PC3 prostate cancer xenografts were grown in nude mice until the tumors reached ~50 mm^3^. PC3 xenografts were irradiated using the Small Animal Radiation Research Platform (SARRP) in the Research Division of Radiation Oncology Department of the Perelman Center for Advanced Medicine at the University of Pennsylvania. Tumor growth delay was measured in four cohorts of 5 matched animals/tumors, each sham irradiated with 2 GY, 4 GY, 8 GY, and 16 GY doses to determine the dose-response curve. Based on this curve, the 4 GY dose was selected as it has sufficient dynamic range to demonstrate response changes with LND. The radiation dose was applied 40 min after LND injection to achieve the most significant acidification and de-energization based on ^31^P MRS results. During the first five days post-radiation, tumor dimensions by calipers and body weight were measured daily. After these first five days, the measurements were made every other day. Tumor volume was calculated from its dimensions as described above. Our endpoint for growth delay experiments was four doublings in tumor size (relative to pretreatment volume). Hence, we investigated the smallest measurable tumor size (50 mm^3^) since we did not want our tumors to exceed 1000 mm^3^.

Initially, 5 animals were allocated to the control and LND alone groups and 9 to each of the 4 GY and LND + 4 GY groups. Mean tumor volumes at day 0 and each group’s standard deviation (S.D.) were reported. The data were explored graphically using local smoothing. Within each of the 4 GY and LND + 4 GY groups, two animals were euthanized at days 12 and 19 post-treatment and were thus excluded from the analysis of the tumor growth data. Before euthanasia, tumor volumes in these animals were similar to other animals in their treatment group. The LND + 4 GY animals were perhaps slightly more prominent. The primary analysis considered growth delay and doubling time. To determine growth delay, we first fit a log-linear model to the four days of tumor volume data up to and including the day of treatment; using the results of this fit, we estimated a ‘baseline’ tumor volume at day 0 of the experiment. We then determined the time needed for each animal to achieve a target volume of three doublings beyond the baseline volume. Several individuals in the 4 GY treated group showed an initial increase in tumor volume, which plateaued and then declined before regrowth again became steady. For the growth delay analysis, we thus omitted any of the volume data up until day 21 for the 4 GY group. Additionally, one animal in the control group significantly declined tumor volume on day 17 of the experiment after achieving a tumor volume of 88% of the target volume on day 14. For this control animal, we interpolated outside the range of the observed volume data to determine a time to three doublings. We determined each animal’s tumor doubling time during the active regrowth, assuming a log-linear model. To make this determination, we chose periods when the growth curves followed a log-linear pattern for most animals, specifically between days 5 and 14 for the control and LND groups and the data between days 35 and 45 for the 4 GY and the LND + 4 GY groups. Using the later time period, we went beyond the time we observed plateaus and/or declines in the growth curves for the 4 GY animals.

### 2.7. Statistical Analysis 

ANOVA with Bonferroni adjustment for multiple comparisons was used to analyze lactate, pHi, pHe, and βNTP/Pi data using SPSS 20 (IBM Corp., Armonk, NY, USA). This approach allowed us to consider the influence of repeated measurements of each replicate over time. Two-sample *t*-tests were used for the analysis of the Seahorse and YSI data. We conducted a growth delay analysis to estimate the effects of treatment on tumor growth [18]. For each animal, we determined the time to three doublings of tumor volume compared to Day 0 of the experiment. Mean differences in the time to regrowth were determined using a *t*-test. Confidence intervals or the interquartile range (IQR) are shown. Hypothesis testing used a Type I error rate of 0.05 and two-sample *t*-tests to analyze the growth delay. To maximize statistical power, we pre-specified the hypothesis tests to be one-sided for the growth delay analysis, as using LND was only of interest in this experiment if it improved tumor response compared to LND + RT. Confidence intervals were two-sided using a 90% level to be consistent with the one-sided hypothesis testing and Type I error rate of 0.05.

We further compared the behavior of the animals in the 4 GY and the LND + 4 GY group during the initial regrowth period between day 0 and either day 34 or day 21. We visualized the data using a local smoothing function (loess) and repeated measures ANOVA to determine whether there were differences in mean tumor volume between groups for days 0 through 35 or days 0 through 21. These analyses were ad hoc in that we chose the times based on data observations. We conducted these analyses in R, Version 3.4.2 R (Foundation for Statistical Computing, Vienna, Austria) [29].

## 3. Results

### 3.1. Seahorse Experiment: Effect of Lonidamine (LND) on Oxygen Consumption Rate (OCR) and Extracellular Acidification Rate (ECAR) of Prostate Cancer Cells

PC3 cells treated with LND had a significantly lower OCR (*p* = 0.001), ECAR (*p* < 0.001), and OCR/ECAR ratio (*p* < 0.001) compared to control PC3 cells (Table 1). Figure 1 shows the OCR as a function of LND dose, demonstrating that LND effectively inhibited the OCR of PC3 cells at concentrations of 100 µM or greater. PC3 cells treated with LND at 1 µM and 10 µM showed little change in OCR compared to the control cells. PC3 cells treated with LND at concentrations greater than 100 µM showed an immediate and sustained drop in OCR that lasted throughout the entirety of the three-hour assay. In comparison, cells treated with LND at concentrations of 1 µM and 10 µM exhibited little to no change over the same time as control cells. LND inhibited maximum OCR at concentrations as low as 1 µM, achieving the most significant inhibition between 100 µM and 200 µM (Figure 1). 

As shown in Figure 1c, LND inhibited ECAR of PC3 cells in a dose-dependent manner immediately after the start of treatment and effectively at concentrations of 100 µM or greater. LND at concentrations of 1 µM and 10 µM had the opposite effect and increased ECAR throughout treatment. LND at concentrations greater than 100 µM induced an acute drop in ECAR below baseline that recovered after 60 min. This is shown in the reduction in ECAR three hours post-LND treatment in Figure 1.

### 3.2. YSI 2300 Stat Plus Measurements: Effect of Lonidamine (LND) on Extracellular Glucose and Lactate Concentrations of Prostate Cancer Cells

PC3 cells treated once with 150 µM LND showed statistically significant differences in glucose consumption and lactate production compared to control PC3 cells (Figure 2). In the control group, glucose concentration in extracellular media decreased from 10.3 ± 0.15 mM at 0 h to 5.03 ± 0.22 mM at 48 h. In the LND group, glucose concentration in extracellular media decreased from 9.96 ± 0.02 mM at 0 h to 6.70 ± 0.15 mM at 48 h. In the control group, lactate concentration in extracellular media increased from 1.70 ± 0.06 mM at 0 h to 13.4 ± 0.24 mM at 48 h. In the LND group, lactate concentration in extracellular media increased from 1.63 ± 0.03 mM at 0 h to 11.2 ± 0.13 mM at 48 h. A single treatment of LND induced a 19.6% reduction in lactate production and a 33.2% reduction in the glucose consumption in the PC3 cells relative to control.

### 3.3. In Vivo ^1^H MRS Experiment: Effect of Lonidamine (LND) on Tumor Lactate Concentration in Prostate Cancer Xenografts

Steady-state levels of tumor lactate (intracellular plus extracellular) were monitored by ^1^H magnetic resonance spectroscopy (MRS) with the Hadamard slice selective multiple quantum coherence (HDMD-Sel-MQC) transfer pulse sequence in tumors of PC3 xenografted mice. The lactate intensity peaked at 80 min post-LND administration with a significance of *p* < 0.001 compared to its baseline integrated intensity (Figure 3). After peaking at 80 min, the lactate signal slowly decreased to baseline values. Note that the decrease in pHi was 0.4 ± 0.1 units, whereas the reduction in pHe was 0.3 ± 0.1 units; since the extracellular volume fraction ranges between about 30 and 50% in experimental tumors (36), which indicates that most of the lactic acid was confined to the intracellular compartment.

### 3.4. In Vivo ^31^P MRS: Effect of Lonidamine (LND) on Tumor pH and Bioenergetics in Prostate Cancer Xenografts 

Intracellular and extracellular tumor pH and bioenergetics were determined via ^31^P MRS in PC3 prostate cancer xenografts grown subcutaneously in athymic nude mice pre- and 180 min post-administration of LND (100 mg/kg, i.p.). LND exhibited a significant, sustained, and tumor-selective decrease in tumor pHi from 6.94 ± 0.02 to 6.49 ± 0.05 (*p* < 0.001), while pHe non-significantly went from 7.0 ± 0.01 to 6.73 ± 0.06 (*p* = 0.11) (Figure 4). A maximum decrease in tumor pHi of 0.4 ± 0.1 unit (*p* < 0.001) was observed at 80 min following LND administration. However, the tumor pHe exhibited a smaller gradual decline by 0.3 ± 0.06 units overall (*p* = 0.11) (Figure 4). Tumor bioenergetics decreased by 75.0 ± 0.12% (*p* = 0.01) relative to the baseline level immediately before LND administration (Figure 4).

### 3.5. Mouse Tumor Growth Delay Studies with Radiation Therapy 

In a growth-delay experiment, LND significantly potentiated the effect of radiation on PC3 cells. An RT dose of 4 GY was based on an initial investigation, suggesting that this dose elicited better tumor response than control or 2 GY and was less aggressive than the 8 or 16 GY doses (Figure 5). Figure 6 shows tumor volumes for the growth delay experiment for individual animals up until 80 days post-treatment, while Figure 7 shows boxplots of the time to regrowth (3 doublings). Table 2 shows the mean difference in growth delay between individual groups. Compared to the control, LND alone yielded a growth delay of 8.7 days (*p* = 0.008), while RT alone yielded a growth delay of 29.8 days (*p* < 0.001), 21.1 days (*p* < 0.001) greater than LND alone (Figure 6 and Figure 7, Table 2). The combination of LND and 4 GY radiation was more effective than either group alone. Growth delay in the LND + 4 GY group was 39.3 days longer than the control (*p* < 0.001), 30.6 days (*p* < 0.001) greater than LND alone, and 9.5 days (*p* = 0.045) more significant than radiation alone (Figure 6, Table 2). 

## 4. Discussion

There is a pressing need to provide more effective and viable non-surgical alternatives to the treatment of prostate cancer. Increasing tumor oxygen levels is one approach to sensitizing hypoxic tumors to radiation therapy [30]. Using similar principles, selective tumor sensitization by lonidamine (LND) followed by radiation therapy could provide curative potential and minimal morbidity [21,30]. LND is known to inhibit tumor oxygen consumption rates (MVO_2_) and modulate cell metabolism in ways that can be combined with other therapies to treat cancer more effectively [17,20,21,22,24,28,30,31,32]. Other properties of LND, such as its solubility in the tumor micro-environment, have also been explored The wide selection of studies carried out on LND makes it a promising candidate as a sensitizer of tumors to radiation therapy.

Our study shows that LND sensitizes PC3 prostate cancer cells to radiation therapy and potentiates the effect of radiation therapy on PC3 mouse xenografts. Tumor growth delay was the greatest in mice treated with 4 GY RT + LND, delaying the growth of tumor cells by an average of 9.5 days compared to radiation alone. These findings support previous studies, which found LND to have antineoplastic properties and showed that LND and radiation therapy could work synergistically to induce apoptosis in cancer cells and be more effective than either therapy alone [33,34]. LND potentiates the effect of radiation therapy mainly through its effects on oxidative phosphorylation. LND potently inhibits the mitochondrial pyruvate carrier (MPC), preventing pyruvate from being carried into the mitochondrial matrix and entering the TCA cycle. Furthermore, LND has been shown to inhibit complex II of the electron transport chain, further stunting TCA cycle activity in treated cells. This decrease in TCA cycle activity results in a reduction of OCR of up to 50% at the 150 μM LND concentration, as indicated by our Seahorse results (Figure 1 and Table 1).

We hypothesize that, by inhibiting the MPC, LND causes tumor cells to consume oxygen at a lower rate, increasing free oxygen availability in the tumor. More free oxygen in the tumor could result in more free oxygen radicals during radiation therapy, significantly increasing the damage done to cancer cells and their DNA. In addition to LND leading to tumor oxygenation and reduced levels of GSH, both of which can lead to radio-sensitization, LND inhibits the repair of radiation-induced potentially lethal damage [21,35,36]. We have shown that LND has minimal to no effect on healthy murine tissues (brain, liver, muscle) [17]. Normal tissue sparing relative to the tumor by LND may occur, because LND is much more effective in tissues/cells that are acutely or chronically acidified before administration of LND, as most cancer cells are [17].

Our ^1^H and ^31^P MRS studies in Figure 3 and Figure 4 show that the administration of LND (100 mg/kg, i.p.) to PC3 prostate cancer xenografts selectively lowers the intracellular tumor pH (pHi) from 6.94 ± 0.02 to 6.49 ± 0.05 (*p* < 0.001), while simultaneously causing a minimal decrease of the extracellular pH (pHe) from 7.00 ± 0.01 to 6.73 ± 0.06 (*p* = 0.11). The bioenergetic status of the PC3 tumor, measured by the (βNTP/Pi) ratio, progressively declines to 75.0 ± 0.12% (*p* = 0.01) of its pretreatment level in 3 h. LND is not a well-documented MCT inhibitor, but studies suggest it is an inhibitor of hexokinase-2 and other glycolytic enzymes and, thus, an inhibitor of glycolysis [22]. However, our data (Figure 3) clearly demonstrate that LND increases steady-state lactate levels in the tumor by about a factor of 2.5 and that this increase occurs predominantly in the intracellular compartment since the pHi decreased much more than pHe and extracellular lactate decreases with time in the presence of LND. While these results appear to contradict the inhibition of the glycolytic metabolism at the level of hexokinase II or phosphofructokinase by LND, we believe the observed intracellular increase of lactate is due to build-up by inhibition of its extrusion from the cell or by other metabolic pathways that could increase lactic acid production, such as glutaminolysis, the malic enzyme, and pyruvate carboxylase, as well as amino acid metabolism. This trapping of lactic acid is also demonstrated in Figure 2, where treatment of PC3 cells with LND decreased lactate concentration in extracellular cell media by almost 20%. Quantitative studies of flux through these pathways are needed to delineate the source of the increased intracellular lactate definitively. The difference in response at the lower pHi would be interesting, because a more significant effect might be seen in tumors than in normal tissue with the introduction of radiation therapy.

It has been reported that acidic environments markedly prolong radiation-induced G2-phase cell cycle arrest in cancer cells [28,31,32]. It has also been suggested that LND interferes with some aspects of the repair of radiation-induced injury; the theoretical basis for that suggestion is the drug’s established inhibitory effects on energy metabolism and the energy-dependent nature of the repair processes [20,21].

Furthermore, ionizing radiation produces a variety of highly reactive free radicals that damage cells, initiate transduction pathways, and alter gene expression. In radiotherapy, the generation of ROS is exploited for cell killing, and a low level of GSH or its depletion by the concomitant use of drugs may increase the radiotherapy response [37]. ROS are potential inducers of apoptosis and programmed cell death because they can activate the mitochondrial apoptosis pathway by direct oxidative damage to DNA [38]. One of our previous studies showed that LND caused a 40% drop in GSH levels at a concentration of 150 μM and induced the production of ROS in DB-1 melanoma cells [21]. Depletion of GSH and ROS production caused by LND can markedly affect radio-sensitivity. In addition to LND leading to tumor oxygenation and reduced levels of GSH, both of which can lead to radio-sensitization, LND can reduce the repair of radiation-induced potentially lethal damage [35,36].

LND has been evaluated in various clinical trials, both as a monotherapy and in combination with other treatments, such as radiation therapy and chemotherapy [39,40,41]. While some studies have shown promising results, especially in combination with radiation therapy, others have reported mixed outcomes or limited efficacy. These studies have revealed wide variations in plasma concentrations of LND among patients [42,43]. The key limitations of LND use are its solubility. LND exhibits limited solubility at neutral pH, which can affect its bioavailability and efficacy, particularly when administered orally [44].

Despite the current limitations and challenges, LND remains a promising candidate for further investigation in cancer therapy. Future research efforts may focus on optimizing treatment protocols and exploring novel combination strategies to overcome resistance and improve patient outcomes. We have shown in this study that one dose of lonidamine can have significant changes on cancer cell metabolism, so the incorporation of multiple doses of LND in future protocols might induce even greater changes in treated cells that would enable a stronger modulation of cancer cell metabolism and further enhance therapeutic efficacy through the mechanisms that we have discussed above. Additionally, advances in drug delivery technologies [45] and the formulation of more soluble derivatives of LND by chemical modification [46] could help overcome some of the existing limitations and enhance the therapeutic potential of LND in the context of radiation therapy and beyond.

## 5. Conclusions

In this study, we used various techniques to explore the effect of the metabolic modulator LND on PC3 prostate cancer cells. The results of our metabolic assays provided by the Seahorse XFe96 and YSI 2300 STAT Plus analyzers showed that LND causes de-energization and acidification of prostate cancer cells through the intracellular trapping of lactate. Using the Seahorse instrument, we demonstrated that LND inhibits respiration, and we suggest that this, in turn, increases tumor oxygenation. Our ^1^H and ^31^P MRS studies on PC3 prostate cancer mouse xenografts further supported these results, which showed an increase in tumor lactate and de-energization through a decreasing tumor βNTP/Pi ratio with LND treatment. A radiation dose-response experiment on PC3 mouse xenografts provided the information needed to select the radiation dose that, while effective, could improve response when combined with LND. With the respiration-inhibiting, de-energizing, and tumor-acidifying properties of LND established and the proper radiation dose selected, a radiation growth delay experiment was carried out on PC3 mouse xenografts. We found that LND successfully sensitized PC3 mouse xenografts to radiation therapy, significantly delaying the growth of tumors compared to radiation or LND alone.

In summary, we have used ^1^H and ^31^P MRS in mouse xenografts and other biochemical techniques to show that the LND metabolic modulation leads to tumor oxygenation that can be a viable strategy for potentiating the effect of radiation therapy on prostate cancer cells. Though further studies must be carried out to continue validation of this approach, our data suggest that developing such combination therapies could benefit prostate cancer patients in the future. These considerations point to the need for further development of potentiation of radiation therapy using metabolic modulators for the eventual management of prostate cancer.

## Figures and Tables

**Figure 1 cancers-16-01384-f001:**
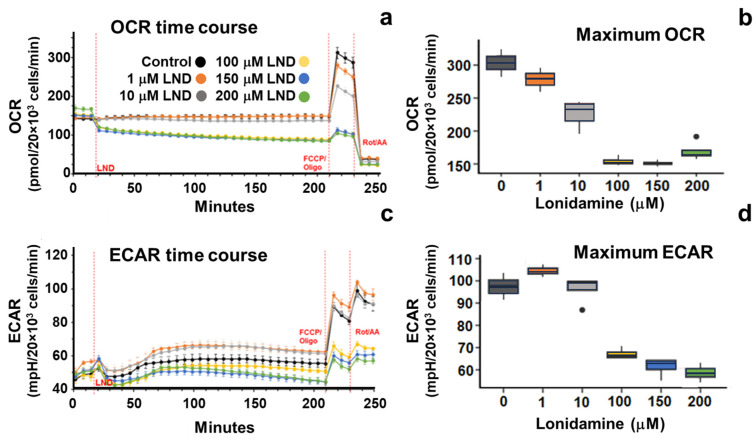
Oxygen consumption rate (OCR) and extracellular acidification rate (ECAR) of PC3 prostate cancer cells determined through the Seahorse XF Cell Energy Phenotype Test using the Seahorse XFe96 Analyzer. Panels (**a**,**c**) show the temporal profiles of OCR and ECAR, respectively. A combination of oligomycin and FCCP induced maximum OCR after treatment with variable doses of lonidamine (LND). Then, a combination of rotenone and antimycin A (Rot/AA) inhibited all mitochondrial respiration. Dashed red lines indicate when the cells were treated with the indicated agents. Data presented as mean ± SEM. Panels (**b**,**d**) show box plots of the dose-dependent response of PC3 cells to LND on the maximum OCR and ECAR. Horizontal lines are the median, while boxes and whiskers show the interquartile ranges. Dots outside of interquartile range represent outliers.

**Figure 2 cancers-16-01384-f002:**
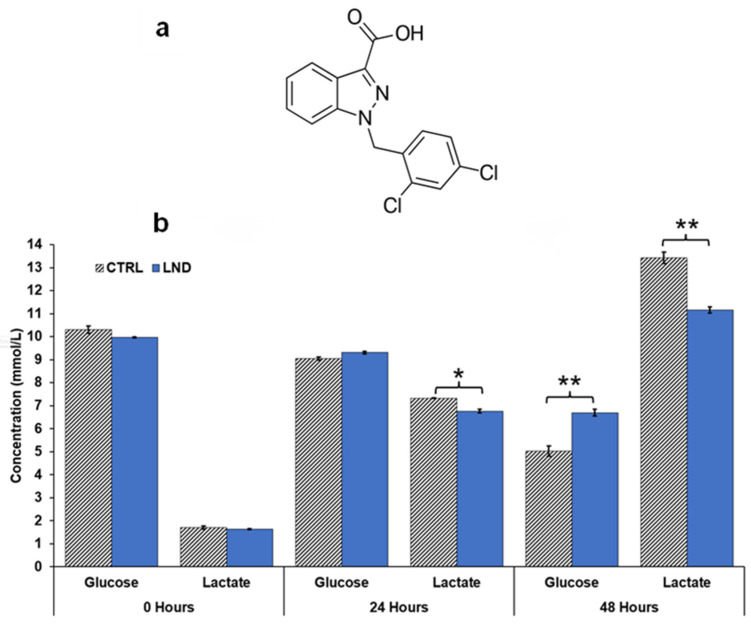
(**a**) Structure of lonidamine (LND). (**b**) Comparison of glucose and lactate concentration changes in extracellular media of PC3 prostate cancer cells under control (CTRL) and 150 µM LND-treated conditions at 0, 24, and 48 h. Metabolic flux was measured using a YSI 2300 STAT Plus Glucose and Lactate Analyzer. Data presented as mean ± S.E.M. (*n* = 3). Statistical significance was determined using a *t*-test. A bracket with one * indicates a significance of *p* < 0.05, while a bracket with two ** indicates a value of *p* < 0.01.

**Figure 3 cancers-16-01384-f003:**
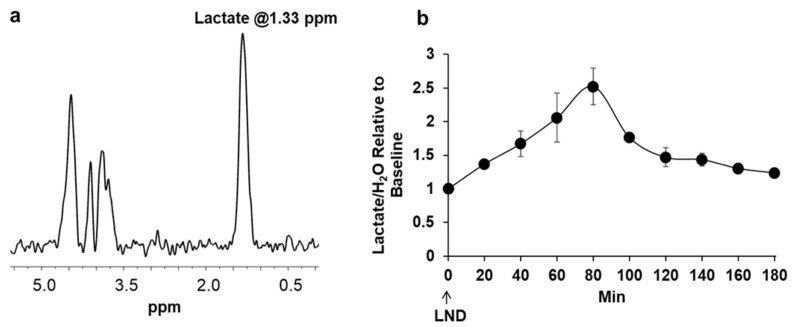
(**a**) Representative ^1^H MRS acquired with the Hadamard Selective Multiple Quantum Coherence transfer pulse sequence (HDMD-Sel-MQC) of PC3 prostate cancer mouse xenografts after administration of lonidamine (LND; 100 mg/kg, i.p.). The lactate peak is at 1.3 ppm, while signals of incomplete water suppression are shown around 4.7 ppm, not affecting the lactate determination. (**b**) Graph showing tumor lactate changes as a function of time in PC3 prostate cancer mouse xenografts after administration of LND. The area under the curve of lactate was normalized to the water signal as described and scaled to the baseline value (time 0).

**Figure 4 cancers-16-01384-f004:**
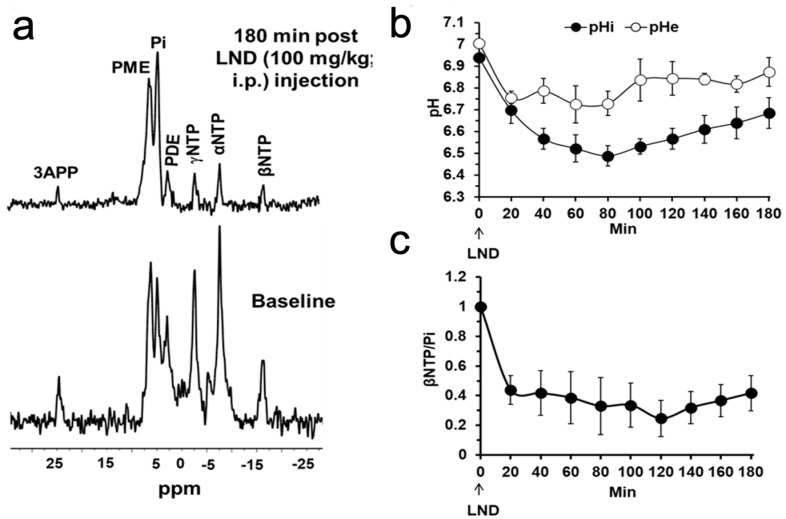
(**a**) Representative in vivo localized (Image Selected In vivo Spectroscopy—ISIS) ^31^P magnetic resonance spectroscopy (MRS) spectra of a PC3 prostate cancer xenograft grown subcutaneously in nude mice (upper) pre- and (lower) 180 min post-administration of lonidamine (LND; 100 mg/kg, i.p.). Resonance assignments are as follows: 3-APP, 3aminopropylphosphonate; PME, phosphomonoesters; Pi, inorganic phosphate; PDE, phosphodiesters; γNTP, γnucleoside triphosphates; αNTP, α-nucleoside triphosphates; βNTP, β-nucleoside triphosphates. (**b**) temporal changes of pHi and pHe after LND. (**c**) Temporal changes of the βNTP/Pi ratio following LND administration indicate impaired energy metabolism.

**Figure 5 cancers-16-01384-f005:**
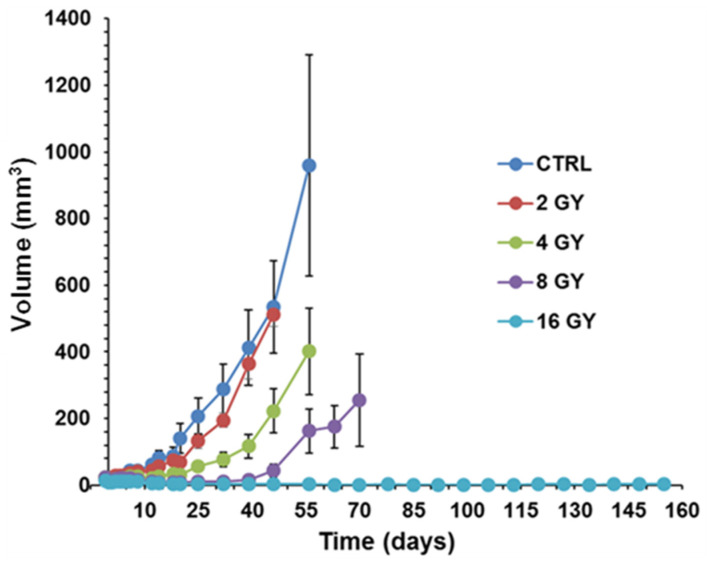
Radiation dose-response curves of a growth delay experiment were used to determine the effective radiation dose for PC3 prostate cancer mouse xenografts. The 4 Gray (GY) dose was selected for further studies because it provided the ideal dynamic range for combination treatment with lonidamine (LND). *n* = 5 for all groups. Values shown are means ± S.E.M.

**Figure 6 cancers-16-01384-f006:**
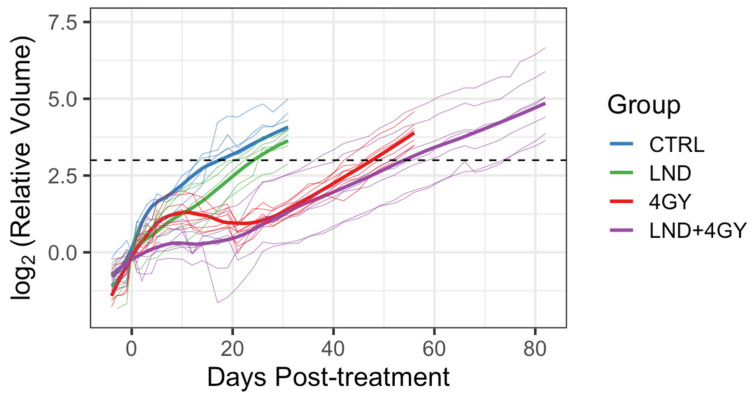
Trajectories of individual mouse tumor volumes post-treatment for a growth delay experiment combining lonidamine (LND) with 4 Gray (GY) of radiation therapy (RT) to enhance the therapeutic effect. Thin lines are tumor volumes for individual animals; darker lines are loess-smoothed data for each group. The dashed horizontal line is 3 doublings relative to the initial volume, defined as ‘tumor regrowth’. Combining LND with 4 GY of radiation significantly enhances the therapeutic effect on PC3 prostate cancer mouse xenografts and was more effective than either LND or RT alone.

**Figure 7 cancers-16-01384-f007:**
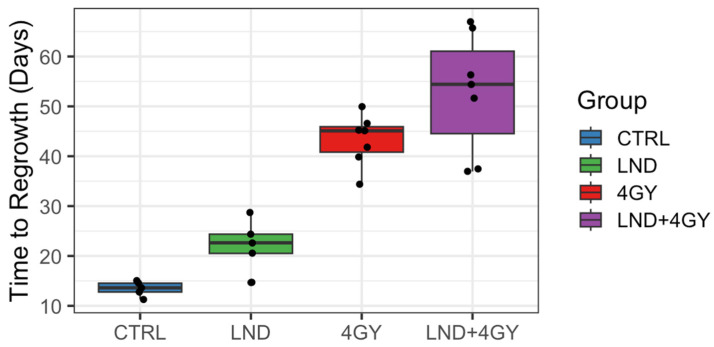
Boxplot of time to regrowth (3 doublings) by treatment group. Horizontal lines are the median, while boxes and whiskers show the interquartile ranges. Dots represent calculated values for each tumor in each group, and those outside the whiskers depict outliers. Sample sizes were 5 for controls (CTRL) and lonidamine (LND) and 7 for 4 Gray (GY) and LND + 4 GY. LND. The time to regrowth mean value of CTRL vs. LND and 4 GY vs. LND + 4 GY statistically differ in a paired analysis (*p* = 0.01 and *p* = 0.02, respectively). The time to regrowth mean value of all the groups statistically differ between each other in a two-sample test when outliers were excluded from the analysis.

**Table 1 cancers-16-01384-t001:** Maximum oxygen consumption rate (OCR) and extracellular acidification rate (ECAR) of PC3 prostate cancer cells were measured using the Seahorse XFe96 Analyzer. Seahorse OCR and ECAR parameters were derived from the Seahorse XF Cell Energy Phenotype Test, performed 3 h post-treatment with 150 µM lonidamine (LND; *n* = 4) or 0.2% DMSO (Control; *n* = 3). Values shown are means ± S.E.M.

Metabolic Parameter	Control	Lonidamine
Basal OCR ^1^	149 ± 6	85 ± 3 *
Stressed OCR ^1^	310 ± 10	113 ± 7 *
Basal ECAR ^2^	55 ± 3	44 ± 2 *
Stressed ECAR ^2^	89 ± 2	60 ± 2 *
Basal OCR/ECAR Ratio ^3^	2.69 ± 0.07	1.92 ± 0.05 *

^1^ pmol/2 × 10^4^ cells/min; ^2^ mpH/2 × 10^4^ cells/min; ^3^ pmol O_2_/mpH; * *p* < 0.05.

**Table 2 cancers-16-01384-t002:** Difference in time to regrowth (days) for pairs of groups. Values shown are mean differences with two-sided 90% confidence intervals. Analysis of PC3 prostate cancer mouse tumor growth delay by treatment with lonidamine (LND) (100 mg/kg; i.p.), radiation therapy (RT) (4 Gray (GY)), and a combination of LND + RT. Comparisons were made between each group to quantify the reduction in tumor regrowth time for radiation therapy by LND. Mice were treated on Day 0 to start the growth delay experiment. The experiment was conducted with *n* = 5 (mice) in the control and LND groups and *n* = 7 for the RT and LND + RT groups.

Group	Control	LND (100 mg/kg; i.p.)	RT (4 GY)
Control	NA		
LND	8.7 (3.8, 13.6), *p* = 0.008		
4 GY	29.8 (26.0, 33.7) *p* < 0.001	21.1 (15.6, 26.6) *p* < 0.001	
LND + 4 GY	39.3 (30.5, 48.2) *p* < 0.001	30.6 (21.2, 40.0) *p* < 0.001	9.5 (0.3, 18.7) *p* = 0.045

## Data Availability

All data are available upon request.

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
