# Peer review of "Lonidamine Induced Selective Acidification and De-Energization of Prostate Cancer Xenografts: Enhanced Tumor Response to Radiation Therapy"

_cancers, 2024, doi:10.3390/cancers16071384_

Round 1

Reviewer 1 Report

Comments and Suggestions for Authors

Thank you for submitting your work to this journal.

First, I suggest you add a paragraph describing the limits of your study and the potential ways of improving your work in the future, maybe by other researchers. The use of radiation, even if it is an established treatment, has its own limits, mainly regarding long term side effects.

Second, as a clinician, I would like to see a paragraph dedicated to current clinical applications of similar techniques, if any, and potential of reducing adverse reactions, e.g. by being able to use smaller amounts of radiation.

Author Response

Reviewer 1

Thank you for submitting your work to this journal.

First, I suggest you add a paragraph describing the limits of your study and the potential ways of improving your work in the future, maybe by other researchers. The use of radiation, even if it is an established treatment, has its own limits, mainly regarding long term side effects.

Second, as a clinician, I would like to see a paragraph dedicated to current clinical applications of similar techniques, if any, and potential of reducing adverse reactions, e.g. by being able to use smaller amounts of radiation.

Response: Thank you for your valuable suggestions. We have included both recommended paragraphs in the Discussion section of the revised manuscript.

Reviewer 2 Report

Comments and Suggestions for Authors

The authors present an interesting study of using Lonidamine to sensitize prostate cancer xenografts to radiation treatment. The resulting effects on oxygen consumption rate, production of lactate and growth delay of tumors are very much worthy of being reported. The in situ NMR experiments are quite interesting as reported. The paper reads well and Figures are clear. Some improvements can be made:

(1) provide a more quantitative discussion of Figure 2 in the text. 

(2) show the structure of lonidamine

Author Response

Reviewer 2

The authors present an interesting study of using Lonidamine to sensitize prostate cancer xenografts to radiation treatment. The resulting effects on oxygen consumption rate, production of lactate and growth delay of tumors are very much worthy of being reported. The in situ NMR experiments are quite interesting as reported. The paper reads well and Figures are clear. Some improvements can be made:

(1) provide a more quantitative discussion of Figure 2 in the text.

(2) show the structure of lonidamine

 Response: Thank you for your appreciation. We have provided the quantitative discussion of figure 2 and structure of lonidamine in the revised manuscript.

Reviewer 3 Report

Comments and Suggestions for Authors

The study by S.Orlovskiy et al. analyzed an important problem in cancer biology and treatment, namely, the role of metabolic processes in prostate cancer cells and their microenvironment in the response to clinical doses of ionizing radiation. The authors used the known drug lonidamine in an attempt to sensitize cells to radiation via metabolic interference. No doubt the problem is pertinent to the readership of Cancers. 

The work is well-designed and comprehensive. The experiments contain proper controls. The statistical analysis is correct.

Two questions are aimed at clarifying the significance of findings:

1. Figure 2: the differences between the groups +/- lonidamine look not too impressive regardless of formal calculations.  Is it correct to interpret these results as stated in lines 339-340?

2. Similarly, the tumor sensitization effect in Figures 6 and 7 (presented as the time to tumor regrowth) is also moderate. May one suggest that one single addition of lonidamine is insufficient? Did the authors try other regimens to increase the efficacy of radiation?  

Author Response

Reviewer 3

The study by S. Orlovskiy et al. analyzed an important problem in cancer biology and treatment, namely, the role of metabolic processes in prostate cancer cells and their microenvironment in the response to clinical doses of ionizing radiation. The authors used the known drug lonidamine in an attempt to sensitize cells to radiation via metabolic interference. No doubt the problem is pertinent to the readership of Cancers.

The work is well-designed and comprehensive. The experiments contain proper controls. The statistical analysis is correct.

Two questions are aimed at clarifying the significance of findings:

  1. Figure 2: the differences between the groups +/- lonidamine look not too impressive regardless of formal calculations. Is it correct to interpret these results as stated in lines 339-340?

  1. Similarly, the tumor sensitization effect in Figures 6 and 7 (presented as the time to tumor regrowth) is also moderate. May one suggest that one single addition of lonidamine is insufficient? Did the authors try other regimens to increase the efficacy of radiation?

Response: Thank you for your suggestions. While the differences in Figure 2 are subtle, these in vitro changes are consistent with what we observe in our in-vivo mouse experiments. We are seeing statistically significant differences in our data that show that LND has the expected effect, which is what we are showing in Figure 2. This data also comes from a single treatment, so treating multiple times might enhance the effect as you have suggested. We have considered your comments about Figure 6 and 7 in this context as well and have made the relevant changes to the manuscript. We have clarified the significance of the findings in the revised manuscript.

Reviewer 4 Report

Comments and Suggestions for Authors

A very good manuscript. Congratulations

I only detected that in Figures 1a and 1c, the units regarding the concentrations tested for lonidamine should be corrected  - replace “uM” by “μM”. Moreover, the quality of the figures could be improved particularly the Figures 1b and 1d.

Author Response

Reviewer 4

A very good manuscript. Congratulations

I only detected that in Figures 1a and 1c, the units regarding the concentrations tested for lonidamine should be corrected - replace “uM” by “μM”. Moreover, the quality of the figures could be improved particularly the Figures 1b and 1d.

Response: Thank you for your appreciation. We apologize for the typos and have corrected them in the revised figures 1a and 1c. We have also replaced the figures 1b and 1d with the improved quality in the revised manuscript.
